# Autophagy and Symbiosis: Membranes, ER, and Speculations

**DOI:** 10.3390/ijms25052918

**Published:** 2024-03-02

**Authors:** Maria G. Semenova, Alekandra N. Petina, Elena E. Fedorova

**Affiliations:** Timiryazev Institute of Plant Physiology, Russian Academy of Science, 127276 Moscow, Russia; masha_semenova99@mail.ru (M.G.S.); sasha.nefedova2@mail.ru (A.N.P.)

**Keywords:** symbiosis, root nodule, autophagy, cell membranes, *Medicago truncatula*

## Abstract

The interaction of plants and soil bacteria rhizobia leads to the formation of root nodule symbiosis. The intracellular form of rhizobia, the symbiosomes, are able to perform the nitrogen fixation by converting atmospheric dinitrogen into ammonia, which is available for plants. The symbiosis involves the resource sharing between two partners, but this exchange does not include equivalence, which can lead to resource scarcity and stress responses of one of the partners. In this review, we analyze the possible involvement of the autophagy pathway in the process of the maintenance of the nitrogen-fixing bacteria intracellular colony and the changes in the endomembrane system of the host cell. According to in silico expression analysis, ATG genes of all groups were expressed in the root nodule, and the expression was developmental zone dependent. The analysis of expression of genes involved in the response to carbon or nitrogen deficiency has shown a suboptimal access to sugars and nitrogen in the nodule tissue. The upregulation of several ER stress genes was also detected. Hence, the root nodule cells are under heavy bacterial infection, carbon deprivation, and insufficient nitrogen supply, making nodule cells prone to autophagy. We speculate that the membrane formation around the intracellular rhizobia may be quite similar to the phagophore formation, and the induction of autophagy and ER stress are essential to the success of this process.

## 1. Introduction

Rhizobia-legume symbioses constitute a quite productive and environmentally friendly nitrogen fixation system, which has a great importance for the global balance of nitrogen [1,2]. Intracellular rhizobia, termed bacteroids or symbiosomes, are able to convert N_2_ from the air into ammonia, the source of nitrogen available for plants [2]. From the nutrition point of view, symbiotic bacteria and apoplast colonies in root nodules belong to biotrophs; however, in senescent and dead cells, they behave as necrotrophs. Rhizobia inhabit the symplast of root nodules in quantities reaching tens of thousands; the intracellular colony lives and functions for quite a long time, up to 6 weeks. Such a situation seems to be rather paradoxical for the eucaryotic plant cells because the elimination of bacteria from the host cells is postponed for a relatively long time. The most intriguing question is: why is the autophagic clearance not induced despite the heavy bacterial infection?

The role of autophagy in symbiosis is not yet sufficiently clarified but has attracted the attention of the scientific community in recent years [3,4,5,6].

In this review, we will discuss the autophagy and its possible connection with the process of establishing and maintaining an intracellular rhizobial colony in the host cell. 

## 2. Autophagy Pathway, Role, and Regulation

Autophagy is the reaction of eukaryotic cells to sugars and nitrogen deficiency, oxygen starvation, heat stress, and pathogen infection [7,8]. Autophagy is characterized by lysis of some organelles and part of the cytoplasm or invading pathogens by the specially formed double-membrane vesicles, autophagosomes. The omegasome, a subdomain of the endoplasmic reticulum (ER), is the source of the autophagosome membrane. Autophagy is a survival mechanism to maintain cellular homeostasis in stress situations such as nutrient depletion or pathogen attack, and it is also involved in programmed cell death [9,10,11,12]. 

Plants have contact with the plethora of microbes that require a constant monitoring and immediate reaction if the pathogen is detected. This is one of the most important reactions of the immune system of eukaryotic cells, which differentiates the “self” and “non-self” and generates effective responses to eliminate the “non-self” agents [10]. The first level of defense involves the induction of transmembrane pattern-recognition receptors of the host plant that distinguish microorganism molecular signatures: microbe-associated molecular patterns (MAMPs) and pathogen-associated molecular patterns (PAMPs) [6]. Commonly occurring bacterial MAMPs include flagellin, surface polysaccharides, lipopolysaccharides, and exopolysaccharides [6,13,14]. 

Plant cells induce a complex defense strategy in response to pathogens. A pattern-triggered immunity reaction includes the production of peroxidases, activated oxygen species, changes in calcium concentration, and modulation of defense through the activity of transcriptional regulators. Defense mechanisms at the level of the plant organism are manifested as stomatal closure, cell-wall strengthening, and the production of antimicrobial compounds [14]. The pathogens are able to retaliate the plant defense reactions and to suppress host immunity by secreting effectors via the bacterial type III secretion system (T3SS) [15] into the plant cell or the apoplast [16]. 

As a part of the response to pathogens, autophagy has been shown to regulate the disease-related cell death response, including the lytic clearance of the whole cell. Such strong immune response is an efficient strategy of an immobile organism, which allows it, at the expense of losing several cells, to restrict further infection [9,10,11,17,18].

Autophagy is also an important housekeeping process that eliminates “not self” invaders, as well as malformed, unwanted, and dysfunctional intracellular components [12,13]. 

## 3. Autophagy-Related Genes and Autophagosome

Most of the autophagy-related genes (ATGs) are conserved in plant genomes and can be divided into five subcomplexes: the ATG1 complex, the class III phosphoinositide 3-kinase (PI3K) complex, the *ATG9* complex, and two ubiquitin-like conjugation systems (*ATG5–ATG12* and *ATG8*) [19].

The formation of autophagosomes starts by the expression of ATG genes and synthesis of the specific proteins that will be recruited to form a developing autophagosome. The process starts with the formation of a bilayer membrane structure that is formed de novo from the subdomain of the endoplasmic reticulum (ER) membrane, positive for PI3P (phosphatidyl-inositol-3-phosphate) and PI3P binding proteins, termed an omegasome. During the induction of the autophagy process, the membranes envelop part of the cell cytoplasm and further extend to form a completely closed membrane-surrounded vesicle, an autophagosome. The PI3P pool engaged in autophagosome biogenesis is synthesized by the class 3 PI3 kinase complex (PI3KC3), comprising VPS34, VPS15, ATG14L, Beclin1, and regulating adaptors, such as VMP1, NRBF2, and Ambra1, and is dependent on ULK1 complex signaling [19,20,21]. Atg1/ULK1 (UNC-51-like autophagy activating kinase 1) initiates the process, and the PI3K (phosphoinositide 3-kinase)/VPS34 (vacuolar protein sorting 34) complex promotes the pre-autophagosome nucleation, adds to the phagophore phosphatidylinositol-3-phosphate (PI3P), and serves as a signal to recruit ATG proteins further downstream [9,22,23,24]. 

The establishment of the contacts of the forming autophagosome membrane with other organelles and with the plasma membrane (PM) is an indispensable part of the autophagosome formation. Such contacts have a central role in the lipid supply, as well as in the membrane expansion [21,23,24,25].

Several types of autophagy are observed in plant cells, including microautophagy, macroautophagy, and mega-autophagy. During microautophagy, the small autophagosomes are transported and fused to the vacuole. Part of the cytoplasm and some cell organelles can be subjected to elimination by macroautophagy, performed by autolysosomes, which contain lytical enzymes. Once the autophagosome fuses with the vacuole, their limiting membrane and contents are degraded by vacuolar hydrolases active against lipids, proteins, nucleic acids, and carbohydrates. Caspase-like vacuolar processing enzyme γ (VPE-γ), the prominent protease, activates a number of vacuolar zymogens [13,18]. Another type of autophagy in plant cells, mega-autophagy, is the cell autolysis and degradation during the process of programmed cell death (PCD) [12,13]. For a detailed description of autophagy regulatory processes in plants, we recommend several excellent reviews [10,11,12,13,14].

Two major signaling kinases are known to regulate autophagy, TOR (target-of-rapamycin) and SnRK1 (SNF1-related protein kinase 1). The TOR complex is a sensor of the nutritional status of the cell and at the same time a negative regulator of autophagy. It consists of the kinase TOR, RAPTOR (regulatory-associated protein of TOR) that promotes complex stabilization and substrate recognition, and the regulatory subunit LST8 [13,14].

Despite that fact that the main source of membrane for autophagosomes is ER, a significant number of ATG genes are responsible for specific lysis of the ER, as well as lysis of mitochondria, plasmids, and the nuclei [19]. The role of some ATG genes has been described [26,27]. The ATG9 gene plays a pivotal role in ER-derived autophagosome formation in plants [28]. ATG2 has been reported to mediate the direct lipid transfer between membranes involved in autophagosome formation [29]. ATG9 and ATG18a play a major role in the autophagosome progression from the ER [15,19]. Autophagosome biogenesis also recruits the actin-related protein 2/3 (ARP2/3) that belongs to an actin nucleation complex. It is required for autophagy, and its trafficking and retrieval system involves ATG2 and ATG18 [12,30,31].

It is also possible that ATG genes may be functional in pathways not related to autophagy. A recent publication by Elander et al. [32], where the authors present the first interactome of Arabidopsis ATG5, hints that plant ATG5 complex proteins have roles beyond autophagy itself. 

## 4. Defense Mechanisms in Symbiosis

Similar to other plant organisms, legumes have an efficient immune system equipped with highly specific mechanisms to recognize a range of microbes, both pathogens and symbionts, and to respond accordingly. The host plant response to Nod factors of symbionts versus the molecular patterns of pathogens reveals two levels of plant responses that include the induction of transmembrane recognition receptors on the plant cell surface. Bacterial flagellin acts as a potent MAMP in many reported plant–pathogen interactions where it is recognized by FLS2, a leucine-rich (LRR) receptor kinase [33]. 

Kouchi et al. [34] have reported a transient defense reaction for a compatible strain of rhizobia in the root nodule of *Lotus japonicus*, manifested by the expression of several defense-induced genes. However, in the work of Kelly [35], is it shown that a symbiotic transcriptomic response of *L. japonicus* to its compatible symbiont *Mesorhizobium loti R7A* and a spectrum of non-symbiotic bacteria produce distinct transcriptional responses for pathogens and symbionts. 

Defense responses during contact between the host plant and rhizobia, however, play an important role in the formation of symbiosis [36,37]. Some defense molecules synthesized as a reaction to bacterial presence are crucial for symbiosome development and polyploidization. The classical example is the synthesis of nodule-specific cysteine-rich (NCR) defensin-like peptides that have a bactericide effect in vitro. The synthesis of these peptides helps the host plant to maintain control of the bacterial population in infected cells, as it causes the terminal differentiation of symbiosomes [38,39,40].

Rhizobia do not behave as parasites, being in the symplast of the host even if they are not capable of fixing atmospheric nitrogen. After analyzing 80 rhizobial strains with mutations in symbiotic properties, Friesen [41] did not detect any examples of an increase in microsymbiont fitness at the host’s expense.

Rhizobia contain many of the commonly occurring bacterial MAMPs, including flagellin, lipopolysaccharides, polysaccharides, and exopolysaccharides. However, competitive rhizobia are able to modulate host defense mechanisms to successfully penetrate the apoplast of the host plant and enter the symplast space of the host cells [42].

Rhizobial microsymbionts employ T3SS [15] effectors to modulate the plant immune response and to suppress the MAMP-triggered immunity; for example, the NopL effector found in *Sinorhizobium fredii* strains NGR234 or the HH103 effector in *Bradyrhizobium elkanii* USDA61 that repress several genes encoding pathogenesis, related to defense proteins associated with MAMP-triggered immunity. Expression of the NopM effector that contains a novel E3 ubiquitin ligase (NEL) domain has been shown to reduce the generation of a reactive oxygen species (ROS) [43,44,45]. The internalization of rhizobia causes the expression in the nodule of some genes that prevents immediate cell death: *DNF2*, *BacA*, and *SymCRK/RSD*. Among the mechanisms that prevent the death of an infected cell, also indicated was the ability of the bacterial strain that elicited the nodules to perform the nitrogen fixation [42,46].

The data concerning the expression of specific markers of defense (pathogenesis-related protein (PRP)) or senescence (cystein protease) indicate that senescence and immunity appear to be antagonists in root nodules, which was shown in a study of mutants with fix^-^ nodules in pea plants [47].

## 5. Autophagy Genes in Root Nodules

Up to now, 39 ATG genes have been detected in the *M. truncatula* genome by genetic screening by Yang et al. [5]. ATGs are functionally classified into core functional groups, namely the ATG1 kinase complex, PI3K complex, ATG9 recycling complex, and two ubiquitin-like conjugation systems. A legume TOR protein kinase that is involved in the regulation of the autophagic process as a response to starvation has been shown to be essential for symbiosis and nodule development [4]. 

## 6. ATGs and Symbiosis

The root nodule, being a temporary organ that integrates two organisms, may have distinctive features regarding the expression of ATG genes and proteins localization that differ from non-infected cells. Rapid proliferation of microsymbiont bacteria and the cost of intracellular colony maintenance inevitably cause a rapid change in the availability of resources for both partners. These circumstances can lead to the induction of autophagy pathways during the formation and maintenance of an intracellular colony.

Data concerning the localization of ATG proteins in infected cells are quite limited. The comprehensive paper of Quezada-Rodríguez [48] is starting to fill this gap, reporting the comparative analysis of 32 genes in *Phaseolus vulgaris*, 39 genes in *M. truncatula*, and 61 genes in *Glycine max*, homologs of Arabidopsis ATG sequences, and the prediction of subcellular localization of ATG18 homologs. The putative localization of ATG proteins in a *M. truncatula*-infected cell is presented in Figure 1.

However, the formation of autophagosomes in mass in young or mature nodule cells of wt nodules has not been stated by microscopy examination, or published, apart from occasional events of microautophagy in young nodules, when young symbioses are engulfed by the vacuole. The rare case in root nodule tissue, when the noticeable quantity of autophagosomes that were engulfing the host cell cytoplasm and organelles were formed, is the situation in nodule infected cells of the DNF1 mutant, the gene that is encoding a subunit of a signal peptidase complex highly expressed in nodules (Figure 2A,B) [49]. In these Fix^-^ nodules, the autophagic bodies with the host cell cytoplasm and organelles were observed (Figure 2A). 

The presented electron microscopy images were created by Fedorova E. The material has been prepared according to the methods described in [49].

To diagnose the dynamics of autophagy gene expression, we performed in silico analysis of ATG gene expression in nodule developmental zones (Table 1A,B). As a query, we used *Arabidopsis thaliana* genes extracted from available databases of genomic sequences and cDNA sequences (https://www.uniprot.org, accessed on 15 August and 15 November 2023) and data from previous reports [5,48]. Selected protein sequences were used for the search of *M. truncatula* homologs in public bioinformatic resources (https://phytozome.jgi.doe.gov/pz/portal, https://www.ncbi.nlm.nih.gov/ (accessed on 25 February 2024). Expression levels of putative *M. truncatula* gene homologs were evaluated in the nodule developmental zones [50] using the Symbimics portal database (https://iant.toulouse.inra.fr/symbimics (accessed on 16–18 September 2023) [51]. 

The data presented on the Symbimics are obtained on the association between the model legume *Medicago truncatula* and its symbiont *Sinorhizobium meliloti*, hence, integrate host plant genes and bacterial microsymbiont genes. The integrated datapool was created by a sensitive and comprehensive approach based upon oriented high-depth RNA sequencing coupled to laser microdissection of nodule regions. Since the Portal data represent expression data in nodule tissues according to a developmental gradient in root nodule, it is a well-recognized source of root nodule gene expression patterns. Table 1A,B presents the TC numbers and in silico analysis of gene expression in nodule developmental zones. Upregulated genes in zone III (zone of active nitrogen fixation) are listed in Table 1A; the genes with unchanged expression or downregulated in the zone of active nitrogen fixation (zone III) are listed in Table 1B. 

According to the data of the in silico analysis, ATG genes from all groups were expressed in the root nodule. Expression was developmental zone dependent. Most ATGs were upregulated in the nitrogen fixation zone (zone III).

The upregulation of some genes increased from 2 to 4 (*MtATG1a, MtATG1t, MtATG2, MtATG6, MtATG7, MtATG11*) or more (*MtATG1t, MtATG3, MtATG13c, MtATG16b*, *MtATG16c*) genes of groups *MtATG8 (MtATG8 a,e,f,g)*, and *MTG18 (a,c,e,f,g)* showed upregulation increases from 5- to 30-fold. The upregulation of genes involved in autophagosome formation from groups *MtATG1, MtATG2, ATG7, ATG8, ATG9, ATG10, ATG11*, and *ATG12* as well as *ATG13,* reflects the fact that the process of autophagosome formation is already induced in the root nodule apical zone and further develops in the infection zone (zone II) and in zone III. 

The upregulation of the ATG18 group genes was quite interesting. Not all functions of the *ATG18* group are yet clarified; however, it is known that genes of this group are involved in osmotically induced and nitrogen starvation-induced vacuole fragmentation in yeast [52,53]. In root nodules, the fragmentation of the vacuole has been described as one of the stages of vacuole defunctionalization, manifested in the loss of vacuole acidic pH in infected cells [54,55]. However, at the time of publication of these articles, data regarding the putative role of the *ATG18* group in vacuole fragmentation in response to starvation had not been available.

With the aim to clarify the environmental factors in the root nodule that can induce expression of ATGs, we performed in silico analysis of gene expression that can be used as a “tester” to such factors as a carbon or nitrogen deficiency [56,57,58,59,60] (Table 2). 

According to this analysis, TOR, a sensor of carbohydrate deficiency and at the same time a negative regulator of autophagy [4], was upregulated in the nitrogen-fixation zone (Table 2). The *SnF* gene [57], which regulates autophagy in conditions of sugar deficiency, was also overexpressed in the nitrogen-fixation zone. The well-known shift from sugars to dicarboxylic acids as the main carbon source for nitrogen-fixing bacteroids has been recently confirmed by metabolomics analysis [61]. This transition can be an adaptation of nodule cells to a shortage of available sugars for the normal nutrition of bacteria inside the infected cells. It also points to an uneven distribution of carbohydrates between macro- and microsymbiont. The *SOC1* [58] gene was also highly upregulated, especially in the meristem, which also indicates a carbon deficiency. The *Dhh1* gene [59], which optimizes the expression of *ATG1* and *ATG 13* under nitrogen starvation conditions, was expressed in all root nodule zones, including zone III, which indicates a suboptimal supply of nitrogen. A high expression of *NAC1* [60], involved in the response to nitrogen deficiency, was detected in the apical part of the nodule, the meristem. Hence, the expression dynamics of these genes may reflect a suboptimal access to sugars and nitrogen in nodule tissue, making nodule cells prone to autophagy. 

We can assume that the root nodule cells are under heavy bacterial infection, carbon deprivation, and insufficient nitrogen supply, and they are reacting accordingly and inducing the expression of ATGs. However, in young developmental zones and in mature nitrogen-fixing nodule cells of nodules, the autophagic clearance of infected cells was not detected.

## 7. ER and Symbiosome Membrane

The formation of an effective nitrogen-fixing symbiosis is impossible without the reprogramming of the endomembrane mechanism of the infected cell [61]. The main sources for the formation of the symbiotic interface are the plasma membrane, the endoplasmic reticulum (ER), and the Golgi vesicles. Symbiosome membrane contacts and fusion with ER and Golgi vesicles are well documented [62,63,64], although the molecular mechanisms of the contact process, the dynamics of membrane expansion, and the regulatory mechanisms have not yet been studied in depth, as the autophagosome formation. 

## 8. Symbiosome and Autophagosome

In between the formation of two membrane interphases, such as the symbiosome and autophagosome membranes, there are significant similarities, and at the same time, a clear difference. The autophagosome is formed de novo from rough ER and keeps the volume according to the volume of the structure, selected for elimination (Figure 2A,B). The Golgi apparatus, endosomes, mitochondria, and the plasma membrane participate directly, indirectly, or partially, in autophagosome biogenesis [24,25,26]. The membrane of the autophagosome definitely belongs to the endocytosis pathway, whose destination is to be fused with the tonoplast.

The symbiosome membrane is not formed de novo, since rhizobia are entering the host cell already enveloped by a membrane that is derived from the host cell plasma membrane [62,63,64]. The symbiosome membrane grows rapidly, and its growth is ensured by the fusion with ER and post-Golgi vesicles; due to this, the symbiosome obtains more lipids and accepts the proteins that belong both to the exo- and endocytotic pathways [65,66]. As a result, the symbiosome membrane has a mixed identity that is shifted from the plasma membrane at the early stage of development to the identity of the early-late endosome/tonoplast during maturation and senescence [64,65,66]. The lifetime of the autophagosome and the symbiosome differ significantly; it is hours in the former and weeks in the latter. Nevertheless, the process of lysis is finally induced in both membrane vesicles; only in the case of symbiosomes, the process is inhibited for quite a long time, up to 3–4 weeks. 

## 9. ER Stress and Membranes

ER is involved in the creation of both autophagosome and symbiosome membranes (Figure 1); therefore, the presence of an extensive ER network may give an advantage and determine the speed of the process of membrane formation. 

The ER is a network of interconnected tubes and flattened vesicles with a high area-to-volume ratio that is situated in the cell cytoplasm [67]. The ER is the main cellular compartment for biosynthesis of most transmembrane and secreted proteins. However, diverse environmental and physiological stress conditions perturb these processes, causing accumulation of damaged or misfolded proteins in the ER lumen and changing the spatial configuration as well as the structure of the ER. This reaction is termed the ER stress. Such conditions cause the activating of the unfolded protein response (UPR) by upregulating protein folding and degradation pathways and inducing autophagy [68,69]. 

The turnover of the ER membrane and its contents is mediated by autophagy, which contributes to ER stress recovery. To alleviate ER stress, the unfolded protein response (UPR) is activated to refold proteins by upregulating the protein-folding machinery and degradative capacity of the ER, allowing plant development. UPR is triggered upon ER stress by the ER transmembrane sensor inositol-requiring enzyme (IRE1). IRE1 senses ER stress through its ER luminal sensing domain and triggers the UPR responses [70,71,72,73,74].

The conditions in root nodule cells, as it seems, are suboptimal. Such situations may produce a large number of improperly folded proteins and create an ER stress reaction. To diagnose the presence of ER stress reactions in the root nodule developmental zones, we have selected marker genes of ER stress (Table 3). 

The in silico analysis of gene expression in nodule developmental zones has shown the upregulation of several ER stress genes (Table 3).

A fairly high level of overexpression in interzone II/III and zone III has been detected for the ER stress response gene Ire1 [74,75] and highly expressed Bag7 [75], BIP2 [76], BCL-2 [75], and Hmg1p [72] (putative *M. truncatula* Karmella gene).

The structural changes in the ER morphology caused by ER stress have been documented, including the increase in ER volume that has been observed as a response to ER stress [72,73]. An E3 ligase, the ubiquitin-fold modifier 1 (Ufm1) ligase 1 (Ufl1), and its small modifier protein Ufm1, as interactors of the core autophagy-related (ATG) proteins ATG1, ATG6, and ATG8, are involved in the ER stress reaction. Ufmylation system mutants have been shown to be hypersensitive to salt stress and trigger the upregulation of endoplasmic reticulum (ER) stress-responsive genes, as well as the accumulation of ER sheets caused by a defect in reticulophagy [72]. The overexpression of certain ER membrane-resident proteins, such as 3-hydroxy-3-methylglutaryl (HMG)-CoA reductase 1 (Hmp1p), also causes the ER stress reported by Zhang et al. [77] and creates the phenotype called “karmellas” that is characterized by the overproduction of ER sheet membranes. In plants, the 3-hydroxy-3-methylglutaryl coenzyme A reductase (HMGR) suffices to trigger ER proliferation [77].

In root nodules, the abundant ER sheets in young infected cells of root nodules were detected more than 30 years ago and have been well described [62,63]. This is a typical phenotype of young infected cells (Figure 3A) that is quite morphologically similar to “karmellas” [78,79].

The presented electron microscopy image was created by Fedorova E. The material has been prepared according to the methods described in [51].

Up to now, the link between the ER stress, autophagy, overproduction of ER membrane sheets, and rhizobia proliferation in infected cells is elusive. 

However, the suboptimal conditions in the nodule, such as deficiency of sugars, nitrogen, and hypoxia, may produce a large number of improperly folded proteins and cause the induction of authophagy genes, as well as ER stress and, putatively, the overproduction of ER membrane sheets [73,76,77,78,79,80]. 

## 10. ER Stress and Autophagy in Symbiosis: The Existing Pathway in a New “Guise”

Very significant membrane resources are required to build a huge symbiotic interface, the symbiosome membrane. Hence, the EP stress, causal for the formation of additional ER membranes in infected cells, may putatively improve membrane resource accessibility for the symbiosome membrane. However, in order for these resources to be utilized in symbiosome membrane formation, mechanisms that regulate contacts, bending and fusion, delivery of necessary molecules and energy resources also need to be induced. It is hard to imagine that these processes would not utilize the machinery already present in the cell, fine-tuned over millions of years, the omnipresent machinery of the autophagy. Perhaps the process is partially modified for a particular purpose, but it is undoubtedly induced in root nodule tissue as a response to an insufficient supply of sugars and nitrogen, hypoxia, and possibly other factors we cannot yet diagnose. However, the induction is strictly necessary for the symbiosis progression. The negative effect of TOR downregulation [4] on nodule growth and symbiosome formation shows a link to the importance of the induction of the autophagy pathway. A signal peptidase complex encoded by the DNF1 [49] gene may be one of the important keys preventing autophagy in infected cells because the mutation of this gene induces the formation of autophagosomes that are involved in lysis of host cell cytoplasm in infected cells, which is not observed in wild-type nodules (Figure 2).

We have repeatedly reiterated in this review that the formation of “typical” autophagosomes was not observed in infected nodule cells. However, the symbiosome might well be an “atypical” autophagosome with partly inhibited autolytical mechanisms. The process of membrane formation around the intracellular rhizobia may be quite similar to the mechanisms of a phagophore formation. These processes deserve further research, at least on the localization of ATGs in infected cells. In this review, we made an attempt to link the process of symbiosis development, symbiosome membrane formation, and the autophagy induced in the nodule. So far, we do not have the amount of data that would allow us to formulate a complete hypothesis. But we hope that this review will help to attract the attention of researchers in the field of symbiosis to substantiate this topic.

## 11. Conclusions

The process of autophagy is one of the important housekeeping processes. For a eukaryote cell, it is one of the ways to repair and revitalize itself, as it allows it to remove non-functional proteins and defend against invading pathogens. Autophagy is caused by stresses as well as by contacts with the bacterial and fungal invaders. Autophagy helps to eliminate the invaders in the shortest time. The residence of thousands of rhizobia in plant cells for more than a month represents an extremely rare situation, and so far, we have not figured out why this happens. What exactly is “turned off” or “turned on” in nodule infected cells? Perhaps, asking the same question, the researchers in the field of symbiosis began to study the ATG genes in the nodule, as we see in several recent publications [3,4,5,6]. The putative link between autophagy and symbiosis, which is the topic of this review, may attract more energy and attention from the audience involved in the study of symbiosis. In terms of practical application of this information, we can point to the possibility of creating a model of symbiosis in non-legume plants.

## 12. Unexplored Mechanisms to Be Studied in Future Research

The selective inhibition of the process of autophagosome formation in the root nodule.The selective mechanisms that postpone the autolytical clearance of symbiosomes.Membrane contacts selective for the ER/symbiosome membrane and for the host plasma membrane.The regulation of protein secretion in contact with symbiosome and autophagosome formation.The mechanisms preventing ER-phagy in infected cells.

## Figures and Tables

**Figure 1 ijms-25-02918-f001:**
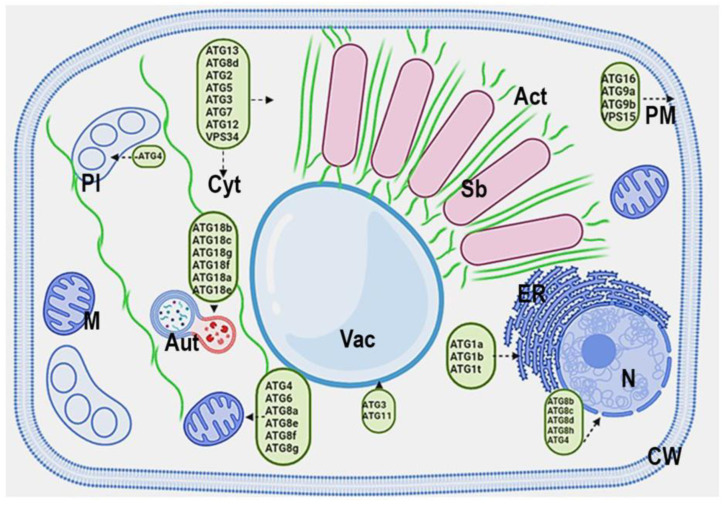
The putative localization of ATG proteins in an infected cell. PM—plasma membrane, Act—actin cytoskeleton, Sb—symbiosome, ER—endoplasmic reticulum, N—nucleus, Vac—vacuole, Aut—autophagosome, PL—plastid, M—mitochondria, Cyt—cytoplasm.

**Figure 2 ijms-25-02918-f002:**
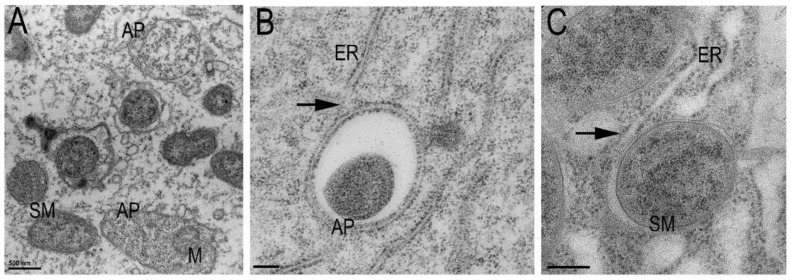
(**A**) DNF1 root nodule, young infected cell. The symbiosomes (SM) and autophagosomes (AP) are present in the same cell. Note part of the host cell cytoplasm and mitochondrion (M) are engulfed by the autophagosome (AP). (**B**) Autophagosome contact with endoplasmic reticulum ER (arrow). (**C**) Symbiosome contact with ER (arrow). Young autophagosome membranes are formed from rough ER; symbiosome membrane is partly free of ribosomes. Note the similarity in morphology of contact of autophagosome (AP) and symbiosome (SM) with ER Bars: (**A**) 500 nm, (**B**,**C**) 200 nm.

**Figure 3 ijms-25-02918-f003:**
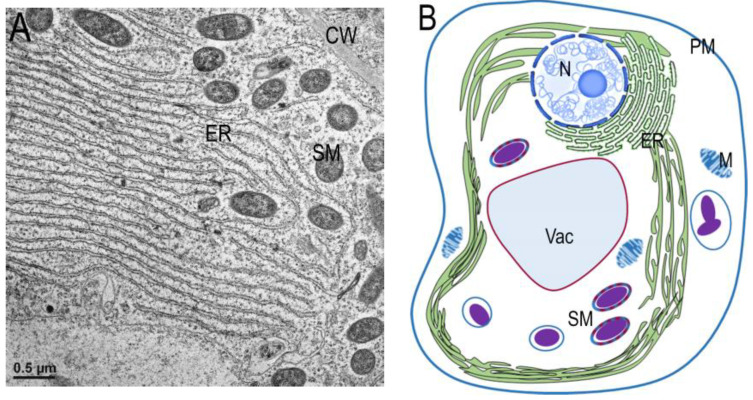
(**A**) Electron microscopy image of the ER sheets, part of a young *M. truncatula* infected cell. (**B**) The scheme of an infected cell. ER—Endoplasmic Reticulum, SM—symbiosome, M—mitochondria, Vac—vacuole, PM—plasma membrane, N-nucleus. Bar (**A**)—0.5 μm.

**Table 1 ijms-25-02918-t001:** (**A**) ATG genes with the expression level maximal in the nitrogen fixation zone (zone ZIII). (**B**) ATG genes with the expression level stable or decreased from apical to nitrogen fixation zone (zone ZIII).

(A)							
Gene Name	Locus ID	Function	Expression (%)				
			FI	FIId	FIIp	IZ	ZIII
*MtATG1a*	*Medtr8g024100*	Autophagosome formation, ER-phagy	13	11	19	30	26
*MtATG1t*	*Medtr3g095620*		4	11	2	12	70
*MtATG2*	*Medtr4g086370*	Lipid transfer, autophagosome membrane termination	13	10	15	27	36
*MtATG3*	*Medtr4g036265*	Cytoplasm-to-vacuole transport	9	14	16	27	34
*MtATG4*	*Medtr7g081230*	Nucleophagy, mitophagy	18	16	13	23	31
*MtATG6*	*Medtr3g018770*	Cytoplasm-to-vacuole transport, nucleo- and mitophagy	14	11	23	18	33
*MtATG7*	*Medtr0003s0540*	ATG12 conjugation with atg5 and atg8	18	14	7	23	38
*MtATG8a*	*Medtr2g023430*	Autophagosome formation, nucleo- and mitophagy	1	7	23	29	40
*MtATG8e,*	*Medtr4g101090*		2	2	3	31	62
*MtATG8f*	*Medtr1g086310*		12	11	8	27	42
*MtATG8g*	*Medtr4g123760*		3	16	21	12	47
*MtATG9a*	*Medtr7g096680*	Autophagosome membrane expansion	6	9	19	47	20
*MtATG9b*	*Medtr1g070160*		25	17	12	21	25
*MtATG11*	*Medtr4g130370*	Scaffold ATG1-ATG13, autophagosome to vacuole delivery	14	15	18	15	38
*MtATG13a*	*Medtr5g068710*	Activation of ATG1 kinase via TOR pathway	14	13	14	29	29
*MtATG13c*	*Medtr8g093050*		1	7	20	29	43
*MtATG16c*	*Medtr4g007500*	Stabilization of ATG5-ATG12 conjugate	7	9	13	27	45
*MtATG18a*	*Medtr1g083230*	The Atg2-Atg18 complex tethers membranes to ER. Osmotically induced vacuole fragmentation in response for starvation.	7	9	13	26	45
*MtATG18c*	*Medtr7g108520*		14	18	10	28	30
*MtATG18e*	*Medtr3g093590*		10	16	11	24	**38**
*MtATG18f*	*Medtr2g082770*		3	3	4	24	**66**
*MtATG18g*	*Medtr1g089110*		7	11	19	23	**40**
(**B**)							
**Gene Name**	**Locus ID**	**Function Symbimics**	**Expression (%)**				
			**FI**	**FIId**	**FIIp**	**IZ**	**ZIII**
*MtATG1b*	*Medtr4g019410*	Autophagosome formation	25	22	8	23	23
*MtATG5*	*Medtr5g076920*	Autophagosome formation	22	26	19	14	19
*MtATG8b*	*Medtr4g037225*	Autophagosome formation, nucleo- and mitophagy	37	0	8	26	29
*MtATG8c*	*Medtr4g048510*		30	27	10	11	22
*MtATG8d*	*Medtr2g088230*		9	15	24	29	23
*MtATG8h*	*Medtr7g096540*		54	30	0	8	7
*MtATG9a*	*Medtr7g096680*	Autophagosome membrane expansion	6	9	19	47	20
*MtATG10*	*Medtr8g010140*	Autophagosome formation	28	27	23	17	6
*MtATG12*	*Medtr8g020500*	Autophagosome formation	14	22	22	23	19
*MtATG13b*	*Medtr3g095570*	Activation of ATG1 kinase, via TOR pathway	4	12	46	17	21
*MtATG16a*	*Medtr3g075400*	Stabilization of the ATG5-ATG12 conjugate	19	54	20	0	7
*MtATG16b*	*Medtr4g104380*		25	19	7	21	27
*MtATG18b*	*Medtr4g130190*	Osmotically induced vacuole fragmentation response for starvation	3	9	22	42	23
*MtATG18h*	*Medtr1g082300*		29	21	12	16	22

**Table 2 ijms-25-02918-t002:** Expression dynamics of autophagy genes responsive for energy, sugars, and nitrogen deficiency.

Gene	Nº NSBI/ Symbimics	Functions	% Expression				
			FI	FIId	FIIp	IZ	ZIII
Genes Whose Expression is Maximal in the Nitrogen Fixation Zone (Zone ZIII)							
*TORC*	NC_053048.1 Mt0026_10261	Central regulators of energy and nutrient perception during plant growth and development	15	14	18	19	33
*Beclin1*	NC_053044.1 Mt0017_00537	Central regulators of energy and nutrient perception during plant growth and development	14	11	23	18	33
*SnF*	NC_053047.1 Mt0025_10284	SnRK1-regulated metabolism and transcription in response to energy starvation and ABA signaling and inactivation by sugars that restore energy balance	13	13	14	23	38
*Dhh1*	NC_053047.1 Mt0025_10284	Dhh1 protein-facilitated translation of ATG1 and ATG13 mRNA under nitrogen starvation conditions	18	15	17	22	24
*SOC1*	Mt0001_01425	A regulator of nutrient starvation whose up-regulation renders the autophagy process more active	55	12	7	12	14
*NAC1*	Mt0003_11375	It plays a role in the response to abiotic stress, a signal for nitrogen deficiency	45	37	0	11	6

**Table 3 ijms-25-02918-t003:** Genes involved in ER stress response in nodule developmental zones.

Gene Name	Locus ID	Functional	Expression, %				
			FI	FIId	FIIp	IZ	ZIII
Ire1	Mt0005_10918	ER stress response	17	15	16	24	26
BIP3	Mt0009_00722	ER stress marker, response to misfolded proteins	20	22	31	19	8
Bag7 (BCL-2-)	Mt0061_10007	Transformation of unfolded protein in response to ER stress	40	34	18	5	2
Hmg1p (Hmg1)	Mt0020_10113 (putative)	Karmella (involved in ER growth) when overexpressed	44	32	12	3	8

## Data Availability

The literature data used in the review are available as publications in relevant scientific journals.

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
