# Peer review of "Autophagy and Symbiosis: Membranes, ER, and Speculations"

_ijms, 2024, doi:10.3390/ijms25052918_

Round 1
Reviewer 1 Report
Comments and Suggestions for Authors
This review is interesting and valuable for readers of IJMS. The similarity is 17% thus it is acceptable. However several points should be improved before proceeding further:
1. The abstract is too general. It needs to summarize the significant findings that they review by detail of results.
2. Discussion part and future research are required.
3. The authors should discuss about shortcomings of researches as they listed and discussed in the research, and add a paragraph what should do in a near future.
4. Many references do not follow the style of MDPI, for instance in the reference 5, the scientific name of the plant was not in italic. Please check carefully other references. Many other references have same problems.
5. Minor English improvement is needed.
I am looking forward to review their revision.
Comments on the Quality of English LanguageGenerally the paper is interesting but some parts need revisions as I indicate
Author Response
The authors are very grateful to the Reviewer for the detailed and professional analysis. We have corrected the abstract, text, conclusions, list of references and the section of Funding and Acknowledgements according to the advice given by the Reviewer.
Rev1
This review is interesting and valuable for readers of IJMS. The similarity is 17% thus it is acceptable. However several points should be improved before proceeding further:
- The abstract is too general. It needs to summarize the significant findings that they review by detail of results.
A: We have rewritten the abstract and have added the part concerning the obtained results ( lines 11-23)
- Discussion part and future research are required.
A: We have discussed the obtained data (lines 411-432)
- The authors should discuss about shortcomings of researches as they listed and discussed in the research, and add a paragraph what should do in a near future.
A: We have expanded the summary of the Review and future development of the topic ( lines 411-432)
- Many references do not follow the style of MDPI, for instance in the reference 5, the scientific name of the plant was not in italic. Please check carefully other references. Many other references have same problems.
A: We have corrected the reference list according to the style of MDPI
Reviewer 2 Report
Comments and Suggestions for Authors
This paper summarizes current knowledge on the roles of autophagy on the interactions of plants and soil bacteria. I believe the authors have done their best in exhausting the topic which could be of interest for specialists in the field.
Content-wise I have no major queries however the manuscript needs a thorough clean-up presentation-wise:
- First of all, please pay close attention to the formatting, unify mentioning the tables and figures in the text (Tab. Versus Table; Fig. Versus Figure). Please re-arrange the tables/figures placing them at the end of paragraph where they are first mentioned.
- Please, check the format of references and revise accordingly.
- Please, add the source of figures. Were they prepared by the authors? If so, how exactly. If not, please add the appropriate source.
- Please, provide a brief description of your in silico analysis for the Tables.
- Please, briefly discuss the importance of autophagy for practical science – why has it attracted the attention of researcher recently?
- I am missing Conclusions, Acknowledgement and Funding section. The review needs a set of concluding remarks – a take home message for the readers.
Comments on the Quality of English LanguageThe authors should revise several grammar errors/typos that appear throughout the manucsript.
Author Response
The authors are very grateful to the Reviewer for the detailed and professional analysis of the Review. We have corrected the abstract, text, conclusions, list of references and the section of Funding and Acknowledgements according to the advices given by the Reviewers.
Comments and Suggestions for Authors
This paper summarizes current knowledge on the roles of autophagy on the interactions of plants and soil bacteria. I believe the authors have done their best in exhausting the topic which could be of interest for specialists in the field.
Content-wise I have no major queries however the manuscript needs a thorough clean-up presentation-wise:
- First of all, please pay close attention to the formatting, unify mentioning the tables and figures in the text (Tab. Versus Table; Fig. Versus Figure). Please re-arrange the tables/figures placing them at the end of paragraph where they are first mentioned.
A: We have corrected the abbreviations and rearranged the tables and figures.
- Please, check the format of references and revise accordingly.
A: We have corrected the reference list according to the suggestions of Reviewers
- Please, add the source of figures. Were they prepared by the authors? If so, how exactly. If not, please add the appropriate source.
A: The reference for the methods and the source of figures are included (lines 223, 224, 385-386)
- Please, provide a brief description of your in silico analysis for the Tables.
A: We have added the description of analysis ( lines 233-242)
- Please, briefly discuss the importance of autophagy for practical science – why has it attracted the attention of researcher recently?
A:We have added the paragraph describing the importance of the topic and future perspectives (lines 416-432)
- I am missing Conclusions, Acknowledgement and Funding section. The review needs a set of concluding remarks – a take home message for the readers.
A: Conclusion is added ( lines 421-432). Acknowledgement and Funding section are added (lines 442-445)